# *MYBPC3* D389V Variant Induces Hypercontractility in Cardiac Organoids

**DOI:** 10.3390/cells13221913

**Published:** 2024-11-19

**Authors:** Darshini Desai, Taejeong Song, Rohit R. Singh, Akhil Baby, James McNamara, Lisa C. Green, Pooneh Nabavizadeh, Mark Ericksen, Sholeh Bazrafshan, Sankar Natesan, Sakthivel Sadayappan

**Affiliations:** 1Center for Cardiovascular Research, Division of Cardiovascular Health and Disease, Department of Internal Medicine, University of Cincinnati College of Medicine, Cincinnati, OH 45267, USA; tjsong@arizona.edu (T.S.); rohit.r.singh@hotmail.com (R.R.S.); akhilbaby@arizona.edu (A.B.); james.mcnamara@mcri.edu.au (J.M.); nabaviph@ucmail.uc.edu (P.N.);; 2Department of Genetic Engineering, School of Biotechnology, Madurai Kamaraj University, Madurai 625021, India

**Keywords:** hypertrophic cardiomyopathy, hypercontraction, MYBPC3, cardiac organoids, mavacamten

## Abstract

*MYBPC3*, encoding cardiac myosin binding protein-C (cMyBP-C), is the most mutated gene known to cause hypertrophic cardiomyopathy (HCM). However, since little is known about the underlying etiology, additional in vitro studies are crucial to defining the underlying molecular mechanisms. Accordingly, this study aimed to investigate the molecular mechanisms underlying the pathogenesis of HCM associated with a polymorphic variant (D389V) in *MYBPC3* by using isogenic human-induced pluripotent stem cell (hiPSC)-derived cardiac organoids (hCOs). The hiPSC-derived cardiomyocytes (hiPSC-CMs) and hCOs were generated from human subjects to define the molecular, cellular, functional, and energetic changes caused by the *MYBPC3*^D389V^ variant, which is associated with increased fractional shortening and highly prevalent in South Asian descendants. Recombinant C0-C2, N’ region of cMyBP-C (wild-type and D389V), and myosin S2 proteins were also utilized to perform binding and motility assays in vitro. Confocal and electron microscopic analyses of hCOs generated from noncarriers (NC) and carriers of the *MYBPC3*^D389V^ variant revealed the presence of highly organized sarcomeres. Furthermore, functional experiments showed hypercontractility, faster calcium cycling, and faster contractile kinetics in hCOs expressing *MYBPC3*^D389V^ than NC hCOs. Interestingly, significantly increased cMyBP-C phosphorylation in *MYBPC3*^D389V^ hCOs was observed, but without changes in total protein levels, in addition to higher oxidative stress and lower mitochondrial membrane potential (ΔΨm). Next, spatial mapping revealed the presence of endothelial cells, fibroblasts, macrophages, immune cells, and cardiomyocytes in the hCOs. The hypercontractile function was significantly improved after the treatment of the myosin inhibitor mavacamten (CAMZYOS^®^) in *MYBPC3*^D389V^ hCOs. Lastly, various vitro binding assays revealed a significant loss of affinity in the presence of *MYBPC3*^D389V^ with myosin S2 region as a likely mechanism for hypercontraction. Conceptually, we showed the feasibility of assessing the functional and molecular mechanisms of HCM using highly translatable hCOs through pragmatic experiments that led to determining the *MYBPC3*^D389V^ hypercontractile phenotype, which was rescued by the administration of a myosin inhibitor.

## 1. Introduction

Hypertrophic cardiomyopathy (HCM) affects as many as 1 in 200 people [1], equating to about 36 million people worldwide [2,3]. HCM is characterized by left ventricular hypertrophy (LVH) with or without left ventricular outflow obstruction. HCM can result in heart failure (HF) with both systolic and diastolic dysfunction [4]. HCM is associated with arrhythmia and sudden cardiac arrest, particularly in young adults and even in trained athletes [5]. Mutation-driven HCM is a primary cause of LVH, a significant indicator of cardiac-related morbidity and mortality [6]. These mutations are common in sarcomeric genes, such as *MYBPC3*, *MYH7*, and *TTN* [7,8,9,10]. However, the mechanism(s) by which these mutations induce LVH is, to date, poorly understood, in turn limiting treatment options for patients harboring these mutations [3,6]. Currently, many clinical and preclinical investigations are underway to define the mechanisms that would explain how sarcomeric gene mutations drive the pathogenesis of LVH in HCM [7]. Elucidation of these mechanisms will lead to the development of targeted interventions. Meanwhile, our group has primarily focused on defining the regulatory role of cardiac myosin binding protein-C (*MYBPC3;* cMyBP-C) in normal cardiac contractile function and examining the consequences of dysregulated contractility caused by genetic mutations [9].

In the sarcomere, cMyBP-C regulates contraction primarily through its N-terminus [11]. Important to the present study are the mechanisms that regulate the interaction between cMyBP-C and the subproximal S2 region of the β-myosin heavy chain [12]. More specifically, cMyBP-C binds to the S2 region of myosin at its C1-M-C2 domains, noting that the interaction is highly regulated by phosphorylation of the M domain [13,14]. Therefore, when cMyBP-C is dephosphorylated, it binds strongly with myosin, thereby inhibiting its force-generating interaction with actin [13,15]. In contrast, when the levels of cMyBP-C phosphorylation increase at serine residues (human cMyBP-C Ser-275, Ser-284, and Ser-304), the interaction with myosin S2 region is ablated, thereby facilitating calcium binding to cardiac troponin C (cTnC) necessary for regular contraction [16]. Therefore, cMyBP-C phosphorylation indirectly regulates calcium binding to the sarcomere through thin filament activation [15,17]. Genetic mutations in cMyBP-C have also resulted in the loss of binding affinity to myosin, causing dysregulation of cardiac contraction and HCM [18]. For example, we previously reported a 25-base-pair (bp) deletion polymorphism (*MYBPC3^Δ25bp^*) found in 6% of those of South Asian ancestry with incomplete penetrance [19]. Moreover, we reported a second variant, *MYBPC3*^D389V^, found exclusively in *MYBPC3^Δ25bp^* carriers [20]. *MYBPC3*^D389V^ was found in 10% of *MYBPC3^Δ25bp^* carriers [20]. Of this subset, the D389V carriers displayed hyperdynamic hearts with 100% penetrance, a common feature of the early onset of HCM [20]. Therefore, defining the pathological regulation of the *MYBPC3*^D389V^ variant will lead to a more comprehensive understanding of HCM pathogenicity.

To this end, we utilized human-induced pluripotent stem cells (hiPSCs) to recapitulate important human developmental stages in vitro and study pathophysiology with high translational value [20]. However, two-dimensional (2D) or monolayer cell models were still far from replicating the structural, functional, and cellular complexity of the tissues and organs they aim to represent [21,22,23]. They lack the complex three-dimensional (3D) matrix, different types of cellular interactions, and the structural orientation of cells [21,22,23]. We, therefore, utilized iPSC-derived cardiac organoid models (hCOs) in the present study [24,25,26]. We have successfully generated highly complex and translationally relevant hCOs using hiPSCs to investigate the regulatory mechanisms that drive *MYBPC3*^D389V^ pathogenicity. Our extensive analyses using hCOs from *MYBPC3*^D389V^ carriers show hypercontractile function, hyperphosphorylation, and oxidative stress. Additionally, we used an FDA-approved myosin inhibitor, mavacamten (CAMZYOS^®^) [27,28], to rescue the hypercontractile phenotype exhibited by the *MYBPC3*^D389V^ variant.

## 2. Methods

A detailed description of experimental materials, methods, and data supporting this study is provided in the Appendix A.

### 2.1. hiPSC Culture and Differentiation into Cardiac Organoids

hiPSCs were cultured in mTeSR™1 plus supplement (Stem Cell Technologies, Vancouver, BC, Canada; Cat. #85850) on a 6-well plate coated with hESC-qualified Matrigel (Corning, New York, NY, USA; Cat #CLS354277) in an incubator at 37 °C and 5% CO_2_ until 80–90% confluency. The hiPSCs are seeded on an agarose mold to form spheroidal organoids, and then differentiation modulates the Wnt pathway [29].

### 2.2. Protein Extraction and Western Blot Analysis

hCOs were processed using urea-based lysis buffer, and proteins were separated in the SDS-PAGE gels (Bio-Rad, Hercules, CA, USA;) and nitrocellulose membranes.

### 2.3. RNAseq Analysis

Total RNA was isolated using the Rneasy Kit (Qiagen, Germantown, MD, USA; Cat # 74034) per the manufacturer’s instructions. Directional polyA RNAseq was performed by the Genomics, Epigenomics and Sequencing Core at the University of Cincinnati as previously published [30,31]

### 2.4. Spatial Molecular Imaging

Spatial molecular imaging was performed on hCO sections by NanoString Technologies Inc. (Seattle, WA, USA) on a CosMx™ Spatial Molecular Imaging (SMI) platform as described previously [32].

### 2.5. Contractility and Calcium Handling

Contractility and calcium handling were measured in beating hCOs using IonOptix system with MyoPacer cell stimulator, HyperSwitch and IonWizard 7.0 acquisition module (Milton, MA, USA) [33]. 

### 2.6. Binding and In Vitro Motility Assays Using Recombinant Proteins

Recombinant human wild-type (WT) and D389V C0-C2 proteins were used to evaluate binding affinity with the proximal myosin S2 peptide by solid-phase binding and co-sedimentation assay, as previously described [12].

To measure the function of actin and myosin filaments in the presence of D389V mutation, we performed an in vitro motility assay using actin filaments sliding over myosin filaments on a 2% dimethyldichlorosilane-coated coverslip as described previously [34].

### 2.7. Statistics and Reproducibility

All analyses were performed using GraphPad Prism 9 software, and all raw data were collected in Microsoft Excel 365 and were normally distributed. Statistical significance was evaluated with a standard unpaired Student *t*-test (2-tailed; *p*  <  0.05) when appropriate (Western blot, ROS generation, membrane potential experiments). For binding and in vitro motility experiments, one-way ANOVA with Tukey’s multiple-comparison test with single pooled variance (*n* = 3, *p*  <  0.05) was used. For multiple-comparison analysis, 2-way ANOVA with post-test correction was applied (*p*  <  0.05). All data are presented as mean  ±  SEM.

## 3. Results

### 3.1. hCOs Generated by Wnt Signaling Modulation

The hiPSCs were generated from peripheral blood mononuclear cells (PBMCs) collected from age-matched noncarrier or D389V carrier human subjects as described previously [20,35]. All hiPSCs were maintained and expanded in mTeSR™1 media. To form 3D aggregates, cells were placed in agarose molds (Figure 1A,B). After they formed a spherical aggregate, media were switched from mTESR1 to RPMI 1640/B27 supplement minus insulin containing 5 µM CHIR99021 for 2 days to activate the canonical Wnt pathway via GSK3 inhibition and addition of IWP2, the inhibitor of Wnt pathway for another 2 days. This is followed by switching the media to RPMI 1640/B27 complete media from day 6 onwards until beating as described in the figure and previous protocols (Figure 1A) [29]. By transmission electron microscopy (TEM), hCOs developed from D389V and NC iPSC lines were shown to possess CM-like ultrastructures containing myofibrils (Figure 1C). Brightfield imaging showed a significant increase in the size of organoids at day 15 of Wnt pathway activation, while confocal microscopy revealed sarcomeric structure upon staining for the *cardiac*-specific markers α-*actinin* (αACT), *cardiac* troponin I (cTnI), and cMyBP-C (Figure 1D–G). Confocal microscopy also revealed the presence of fibroblasts in these hCOs, as detected by vimentin, a well-known marker for fibroblasts [36]. These results demonstrate that hCOs mimic the complex structure of cardiac tissue and are an ideal model for studying normal and pathologic cardiac muscle structure and function in vitro.

### 3.2. D389V hCOs Exhibit Hypercontractility Mitigated by Mavacamten Treatment

Next, we used the IonOptix system to assess the contractile mechanics of hCOs generated from NC and D389V hiPSC lines. hCOs were paced at 0.5 Hz stimulation to minimize inherent beating rate inconsistencies. Contraction velocity was significantly increased in D389V organoids compared to NC organoids at baseline, but relaxation velocity was not different (Figure 2A,B). Also, hCOs derived from D389V carriers showed significantly faster kinetics with time to reach 50% and 70% peak contractions than NC hCOs (Figure 2C,D). To mitigate this hypercontractile phenotype, hCOs were treated with a myosin inhibitor, 300 nM mavacamten (MyK-461, MedChemExpress, Cat. # HY-109037), and contractile properties were assessed post-treatment. As expected, mavacamten reduced the hypercontractility exhibited by D389V hCOs, as evidenced by the decreasing the contraction velocity and increasing time to contract by 50% and 70% (Figure 2A,C,D), while it had no significant effect on the contractility of NC hCOs. Interestingly, we observed significantly faster relaxation kinetics with time taken to 90% and 50% relaxation reduced in D389V organoids (Figure 2E,F). As expected, mavacamten did not affect the relaxation parameter in D389V hCOs (Figure 2E,F). Altogether, these data using hCOs are consistent with our previous findings of hyperdynamic hearts in D389V carriers with increased cardiac function [20].

### 3.3. D389V hCOs Exhibited Faster Calcium Cycling

To examine the effects of altered contractile mechanics on Ca^2+^-handling properties, we measured changes in the amplitude and kinetics of intracellular Ca^2+^ transients in NC and D389V hCOs using fluorescent Fura-2 AM dye (Figure 3A). Our results showed no significant change in Ca^2+^ transient amplitude, expressed as Fura-2 ratio (340/380) and diastolic calcium levels between NC and D389V hCOs at baseline (Figure 3B,C). However, the time to 50% decay was significantly reduced in the D389V hCOs compared to that in NC hCOs (Figure 3D). Interestingly, mavacamten treatment (300 nM) reduced the diastolic calcium, calcium amplitude, and time to 50% decay in D389V hCOs. The findings from D389V hCOs suggest that calcium handling is modified specifically in terms of its removal from the cytosol.

### 3.4. Transcriptomic Analysis Reveals Upregulation of Cardiomyocyte Contraction and Oxidative Phosphorylation in D389V hCOs

We next performed RNAseq analysis to investigate the impact of the D389V mutation on differential RNA expressions in D389V and NC hCOs. Initially, we conducted enrichment testing using upregulated differential gene expressions (DGEs), revealing the top upregulated gene ontology biological processes (GOBP), gene ontology molecular functions (GOMF), and Kyoto Encyclopedia of Genes and Genomes (KEGG) pathways (Figure 4A). The D389V hCOs demonstrated upregulated ontology processes associated with cardiomyocyte functions, including the regulation of contractility and calcium handling, as well as metabolic processes such as oxidative phosphorylation, glucose catabolic process, and the respiratory electron chain (Figure 4A). Gene set enrichment analysis (GSEA) identified 162 differentially regulated KEGG pathways (55 up and 107 down). KEGG pathway analysis showed upregulation of genes responsible for cardiac muscle contraction, hypertrophic cardiomyopathy, oxidative phosphorylation, and glycolysis–gluconeogenesis in the D389V hCOs compared to NC hCOs (Figure 4B–H). Additionally, the hypertrophic genes MYH7 and MYL2 were significantly increased in D389V hCOs. The upregulation of these genes is consistent with early-stage cardiac hypertrophy (Figure 4F). Conversely, D389V hCOs showed downregulation of genes involved in fatty acid metabolism, as illustrated in the heatmap and enrichment plot (Figure 4E,I). Genes encoding calcium, sodium, and potassium channels (e.g., RYR2, CACNA1C, and KCNJ5) were significantly increased in D389V hCOs compared to NC controls (Appendix A). To examine the impact of 3D culture conditions, we compared DGEs between 3D hCOs and 2D hiPSC-CMs from NC hiPSCs (Appendix A). We isolated the upregulated and downregulated genes in both DGE comparisons. Interestingly, compared to 2D hiPSC-CMs, 3D hCOs showed upregulation of genes associated with metabolic processes, blood circulation, and muscle synthetic GOBP (Appendix A). Moreover, 3D hCOs revealed upregulation of genes associated with GOCC (Gene Ontology Cellular Components), such as the sarcomere, extracellular matrix, and cardiac myofibril formation (Appendix A). Interestingly, we observed the downregulation of genes associated with neuronal development in 3D hCOs. Altogether, RNAseq analysis indicates the upregulation of genes involved in cardiac hypertrophy and mitochondrial dysfunction in D389V hCOs.

### 3.5. D389V hCOs Exhibit Increased cMyBP-C Phosphorylation and Oxidative Stress

To verify whether mutant mRNA was transcribed in D389V cardiomyocytes, exon 12 of *MYBPC3* with 390 bp was amplified and gel extracted. The sequencing of the following region did confirm the presence of mutant mRNA of heterozygous A and T peaks observed in D389V cardiomyocytes compared to NC, which only has an A peak (Appendix A). Western blot experiments revealed that protein lysates from hCOs with D389V and NC revealed no differences in total cMyBP-C expression levels; however, phosphorylation of cMyBP-C normalized with total protein was significantly increased at serine 275 and 304 in D389V hCOs compared to that in NC hCOs (Figure 5A–G). Given the translational relevance of this work, we next investigated the metabolic implications of D389V in cardiac organoids. Pathway analysis of differentially expressed genes (DEGs) in D389V hCOs showed upregulation of gene ontology biological processes like triglyceride metabolic process (*p*-value = 0.0094), glucose catabolic process to pyruvate (*p*-value = 0.019), canonical glycolysis (*p*-value = 0.0195), and glycolytic process through glucose-6-phosphate (*p*-value = 0.0227), indicating altered metabolic responses (Appendix A). Next, we measured oxidative stress in hCOs, using the DCFDA dye-based method to detect reactive oxygen species (ROS) levels [37]. Our results showed an increase in ROS in D389V hCOs compared to NC hCOs, indicating increased oxidative stress in mutant carriers (Figure 5H,I), which was resolved by N-Acetyl cysteine (NAC), an antioxidant treatment [38]. We also measured mitochondrial membrane potential, which was found to be lowered in D389V cardiomyocytes compared to NC cardiomyocytes, indicating mitochondrial dysfunction (Figure 5J). The transcriptomic data combined with functional analyses showed that D389V hCOs recapitulate key aspects of metabolism in early cardiac hypertrophic phenotype.

### 3.6. Spatial Mapping Demonstrates the Presence of Enriched Cardiomyocytes

Using the CosMX spatial molecular imager, we then performed spatial mapping of hCOs to profile the various cell types. CosMx Human RNA TAP Panel (1000-plex) and custom panel, consisting of genes that included *SERCA2a*, *MYBPC3*, *MYH6*, *MYH7*, *NPPA*, *TNNI3*, and *TNNT2*, were used to probe the cells in hCOs to quantitate the level of these RNA transcripts. Overall, the spatial profile of hCOs suggests efficient cardiac differentiation based on the expression levels of cardiomyocyte genes and the presence of other cell types, such as endothelial cells and fibroblasts. UMAP projections in both groups of organoids showed similar expressions of cardiomyocyte cell populations, as denoted by clusters d and e (Figure 6A). Interestingly, Uniform Manifold Approximation and Projection (UMAP) of D389V hCOs showed a higher presence of immune cells, as denoted by cluster “f”, than that found in NC hCOs (Figure 6A,B). The heatmap showed different types of gene expression in different cell populations within D389V and NC hCOs. Interestingly, the hypertrophic genes *NPPA* and *NFAT1* were elevated in D389V hCOs in the cardiomyocytes compared to NC hCOs. Higher expression of fibrotic genes, including *COL11A1,* was exclusive to D389V hCOs within fibroblast cell types compared to NC hCOs (Figure 6C).

### 3.7. D389V Dampens the Interaction Between cMyBP-C and Myosin S2 Region

The effect of D389V on the binding interaction between cMyBP-C and myosin S2 region was determined in vitro. We hypothesized that loss of aspartic acid at position 389 would, in turn, result in the loss of cMyBP-C binding to the myosin S2 region. To test the hypothesis, we used co-sedimentation, solid-phase binding, isothermal titrating calorimetry, and in vitro motility assays to test the impact of D389V on the binding of cMyBP-C and myosin S2 region. For the co-sedimentation assay, whole-length cardiac myosin (10 µM) isolated from mice hearts was incubated with increasing concentration (0–10 µM) of wild-type recombinant human cMyBP-C C0-C2 region (hC0-C2^WT^) and D389V (hC0-C2^D389V^) [39]. The relative binding max (B_max_) of hC0-C2^D389V^ (0.38 ± 0.09) was significantly lower (*p* = 0.025) compared to that of mC0-C2^WT^ (0.86 ± 0.1). However, the dissociation constant (*K*_d_) for hC0-C2^D389V^ (0.9 ± 0.31 µM) to myosin remained unchanged (*p* = 0.9368) compared to that of hC0-C2^WT^ (0.84 ± 0.64 µM) (Figure 7A).

Next, the binding of recombinant proximal (126 residues) human myosin S2 (hS2) with hC0-C2^WT^ and hC0-C2^D389V^ was tested by solid-phase binding assay (SPBA). For this assay, 20 µM hC0-C2^WT^ and hC0-C2^D389V^ were incubated with increasing concentrations of hS2 (0–20 µM), and the binding of hS2 to hC0-C2^WT^ or hC0-C2^D389V^ was determined by fluorescence intensity [12]. Again, the relative B_max_ of hC0-C2^D389V^ to hS2 (0.94 ± 0.08) decreased significantly (*p* = 0.04) compared to that of hC0-C2^WT^ (1.21 ± 0.09). Similar to the co-sedimentation assay, hC0-C2^D389V^ (4.73 ± 1.07 µM) had no significant effect (*p* = 0.8795) on *K*_d_ in comparison to that of hC0-C2^WT^ (4.95 ± 0.94 µM) (Figure 7B).

For isothermal calorimetry analysis (ITC), 20 µM of hC0-C2^WT^ or hC0-C2^D389V^ were titrated against 350 µM hS2 proteins. Compared to hC0-C2^WT^, hC0-C2^D389V^ had minimal to no heat change per second (ΔCp). The stoichiometry (η) of hC0-C2^D389V^ to hS2 was out of range at 280.9 ± 453.7, while that of hC0-C2^WT^ to hS2 was at 0.96 ± 0.01. The slope of the reaction was significantly lowered for hC0-C2^D389V^ (0.45 ± 0.14) from that of hC0-C2^WT^ (4.18 ± 0.29), suggesting low to minimal binding of hC0-C2^D389V^ to hS2^WT^ (Figure 7C).

Lastly, we measured the motility of fluorescently labeled F-actin over heavy meromyosin (HMM) of human β-cardiac myosin heavy chain in the presence of hC0-C2^WT^ and hC0-C2^D389V^ in the equimolar concentration of 5 µM. The hC0-C2^WT^ lowered the motility of F-actin over HMM to 7.34 ± 0.34 µm/s from 10.29 ± 0.43 µm/s in the absence of C0-C2. However, when the motility of F-actin was challenged with hC0-C2^D389V^, it had no significant effect (*p* = 0.1173) on the velocity (9.1 ± 0.35 µm/s) when compared to motility in the absence of C0-C2 (Figure 7D). Altogether, the results of these assays suggest that D389V mutation does reduce the binding of cMyBP-C to the myosin S2 region, which is a critical determinant for hypercontraction.

## 4. Discussion

The genetic basis of HCM owing to mutations in sarcomere genes has been studied extensively using various in vitro and in vivo models. However, variations in the clinical presentation of HCM make it necessary to further clarify clinical intervention in genotypic–phenotypic correlated studies and recapitulate the clinical phenotype in experimental models [6,7]. Previously, our group showed that the *MYBPC3*^D389V^ variant in humans was associated with overall hyperdynamic cardiac function [20]. We had previously used iPSC-CMs to show the presence of cellular hypertrophy and irregular calcium handling in the presence of the *MYBPC3*^D389V^ variant [20]. To validate these findings, determine the molecular mechanism of the *MYBPC3*^D389V^ variant, and evaluate small-molecule therapeutics, we make use of highly translational 3D hCO models.

### 4.1. Three-Dimensional Cardiac Organoids Are a Model for HCM

The discovery of iPSCs was a revolutionary tool for the study of human cardiac development and the pathophysiologic mechanisms of cardiac disorders [40,41]. As iPSCs are derived from patients, their use allows precise disease modeling [24,26,42]. Moreover, the latest gene editing approaches have made it possible to study mutations in iPSC-CMs in vitro, providing newer insights into disease mechanisms [43]. Nonetheless, such a 2D culture system has its limitations. Specifically, 2D cultures differ from in situ pathologies in terms of cell morphology, differentiation capability, and gene/protein expression levels. Also, traditional 2D cell cultures lead to pure cell lineages that fail to fully reflect the interaction or crosstalk between cells or with the extracellular matrix [44]. Animal models, on the other hand, are usually limited by species specificity and heterogeneity [45]. These limitations create challenges in identifying successful and biologically relevant translational models that possess cellular complexity. Therefore, 3D hCOs present a promising alternative to traditional 2D culture systems and animal models in the study of cardiac disorders. However, 3D hCOs allow for more complexity, reproducibility, and a closer resemblance to in vivo cell characteristics compared with 2D cultures. As such, 3D hCOs provide a more biologically relevant translational model for studying disease mechanisms and therapeutic interventions [46]. Overall, new models like 3D hCOs that maintain genetic characteristics and functions comparable to those of in vivo cells hold promise. Therefore, in the present study, we made use of the 3D hCO model to understand the pathophysiology associated with the *MYBPC3^D389V^* variant.

An important feature of mature cardiomyocytes is the presence of a distinct ultrastructure of the sarcomere, showing length and alignment, together with the interspacing of myosin–actin [43]. TEM imaging of hCOs in the present study shows highly organized sarcomeres with distinct z-disks. Also, we showed that the 3D hCO model possesses the various cell types needed to recapitulate the human heart in situ. Confocal imaging of hCOs shows the expression of cardiac markers like cTnI, cMyBP-C, and the fibroblast marker vimentin. Moreover, spatial profiling of our hCOs showed the gene expression of various cell markers, including cardiomyocytes, endothelial cells, fibroblasts, immune cells, and even macrophages. We also performed RNA sequencing to compare 2D iPSCs against 3D hCOs. The RNAseq data showed that 3D hCOs had higher gene expression in matrix formation than 2D cardiomyocytes. In addition, the upregulation of genes that participate in ion transport, sarcomere formation, contractility, heart conduction, and other biological processes was observed in the 3D hCOs. Interestingly, the cardiac differentiation efficiency was improved in the 3D system as the iPSC differentiation into non-myocytes was reduced in 3D hCOs compared to 2D culture. For instance, we observed downregulation of gene expression related to neuronal development in 3D systems compared to 2D cultures. These data further confirm that 3D cultures more accurately represent the cellular environment of the myocardium. All these data indicate the advantage of 3D hCOs over 2D iPSC-CM as a translationally relevant cardiac model.

### 4.2. D389V Variant Expression in hCOs Results in Hypercontraction

Mutations in cMyBP-C are the most frequent cause of HCM disorders, comprising 40–50% HCM-associated mutations and resulting in either haploinsufficiency or poison polypeptides [47,48]. Haploinsufficiency and knockdown of cMyBP-C lead to enhanced actomyosin interactions, resulting in increased cross-bridge formation force generation and, ultimately, HCM [48,49]. Previously, our lab and others have shown the significance of cMyBP-C expression and phosphorylation in the background of cardiac physiology and function [50,51,52]. Three phosphorylation sites in the myosin interacting domain of cMyBP-C are serine 273, 282, and 302 in mice and serine 275, 284, and 304 in humans [53,54]. Upon phosphorylation of these sites, the binding of myosin and cMyBP-C weakens, freeing the myosin heads to interact with actin and increasing cardiomyocyte contraction. Conversely, decreased phosphorylation correlates with more inhibition of the myosin heads and reduced cardiac contraction [50,51,53,55]. Based on these data, we hypothesized two mechanisms for the increased hypercontraction seen in the D389V hCOs. The first involves the loss of aspartic acid and the presence of valine on codon 389 in the myosin binding site, which would constitutively cause the loss of cMyBP-C binding interaction with the myosin S2 region, resulting in increased cross-bridges and contractility. This hypothesis was validated by in vitro experiments, such as co-sedimentation, SPBA, ITC, and in vitro motility assays. It can also be hypothesized that increased hypercontraction results from increased cMyBP-C phosphorylation at Ser275 and Ser304 sites in D389V hCOs, which are protein kinase A, protein kinase D, Ca^2+^-calmodulin-activated kinase II and protein kinase C (PKC) sites [50,51,53,56]. Similar phenomena were also observed in the leptin receptor-deficient mouse model, where the hypercontractile heart displayed increased cMyBP-C phosphorylation at Ser273 and Ser302 [57]. Interestingly, RNAseq analysis showed an increase in the gene expression levels of protein kinase C alpha (PRKCA) in our D389V hCOs compared to NC hCOs (Appendix A). PKC alpha is the predominant isoform of PKC expressed in the cardiomyocytes [58]. Taken together, such differential phosphorylation suggests that the kinase landscape might have been altered by the increased phosphorylation level of cMyBP-C in D389V hCOs.

Next, we also analyzed the functional consequences of the D389V variant. Our results revealed that hCOs carrying the D389V showed increased contraction velocity with faster contractile and relaxation kinetics since the time taken to contract 50%, 70%, and time taken to relax 90% and 50% was reduced compared to WT hCOs. At the same time, however, we saw no significant changes in calcium amplitude and diastolic calcium between NC and D389V hCOs. However, a faster decay rate of calcium was observed, indicating increased cellular calcium extrusion from the cytosol. The faster uptake of calcium could be a compensatory response to hypercontractility, ensuring enough calcium availability for the subsequent depolarization cycle. A similar phenomenon was also observed in myocytes from homozygous *MYBPC3* C.772G>A mouse hearts, a mutation that also results in the replacement of a negatively charged amino acid (Glu258Lys) in which faster decay, but no significant change in amplitude, was observed [59,60]. The lack of change in calcium amplitude observed in the D389V hCOs could also indicate that the observed hypercontractility likely results from increased actomyosin interaction.

### 4.3. D389V Variant in hCOs Results in Cellular Hypertrophy

RNAseq and spatial profiling data further supported the presence of cellular hypertrophy in D389V hCOs, including upregulation of *NPPA*, *MYH7*, *TNNI3*, *MYL2*, *TTN*, and *MYBPC3* genes, compared to NC hCOs. Moreover, the upregulation of genes associated with the oxidative phosphorylation pathway was also observed in the D389V hCOs. As previously described [61], an increase in the oxidative pathway is associated with oxidative stress. Thus, we believed oxidative stress might be a secondary response to hypercontractile myofilaments, a theory supported by the increased ROS generation observed in the D389V hCOs. Based on this ROS overload, reduced mitochondrial membrane potential, which is an indication of mitochondrial dysfunction [62], could be linked to cellular hypertrophy as another secondary response. Collectively, these results point to oxidative stress and mitochondrial dysfunction as secondary effects of hypercontractile myofilaments that accompany early HCM phenotype [63,64]. Interestingly, spatial profiling data also showed increased expression of immune cells in D389V hCOs compared to NC hCOs with subsequent increased expression of inflammation-related genes like *VCAM* and *SERPINA3* in D389V organoid samples. This phenomenon is often seen in the cardiac remodeling setting, leading to heart failure [65,66].

### 4.4. Mavacamten Improves Hypercontraction with hCOs Expressing D389V

D389V in the C2 domain of the cMyBP-C, which is the myosin S2 binding region, leads to activation of myosin and hypercontraction, as presented in humans [20] and hCOs carrying the D389V variant. Therefore, we hypothesized that treatment with mavacamten, a small-molecule inhibitor of myosin, would diminish myosin function in the D389V sarcomere and thus mitigate hypercontractility. Post-treatment with mavacamten, we did observe a reduction in the heightened contractile parameters of the D389V hCOs. Furthermore, mavacamten treatment also further reduced the diastolic calcium and calcium decay in D389V hCOs. Overall, our data support the long-term goal of using hCOs to model genetic causes of cardiomyopathies and find treatments specific to a patient’s unique genetics.

### 4.5. Limitations of the Study

Although the hCO model offers exciting prospects for modeling HCM in vitro, significant limitations exist. Organoids can divert from their normal developmental pathway over time in culture. Also, their ability to recapitulate adult cardiomyocytes is still limited compared to that of rodent models, which possess a fully developed immune system. Therefore, technological advancement is needed to better recapitulate the human heart morphologically, anatomically, and physiologically, which may be accomplished by adding other cell types like fibroblasts, smooth muscle cells, immune cells, and mature ventricular and atrial endothelial cells when forming the 3D mold. The addition of these cells would also aid in the formation of vascular networks to provide adequate nutrients, as well as sufficient crosstalk, for cell signaling. Another limitation of the study was the inability to efficiently isolate single cardiomyocytes from cardiac organoids using trypsin to measure cell size and cellular hypertrophy. Also, this study was performed using an iPSC line derived from one noncarrier and one D389V subject. Therefore, more experiments are needed using isogenic controls. For instance, the D389V mutation could be introduced using CRISPR, and recovery could be achieved by knocking down the mutation to ensure that a causal relationship does exist between hypercontractile phenotype and D389V mutation.

## Figures and Tables

**Figure 1 cells-13-01913-f001:**
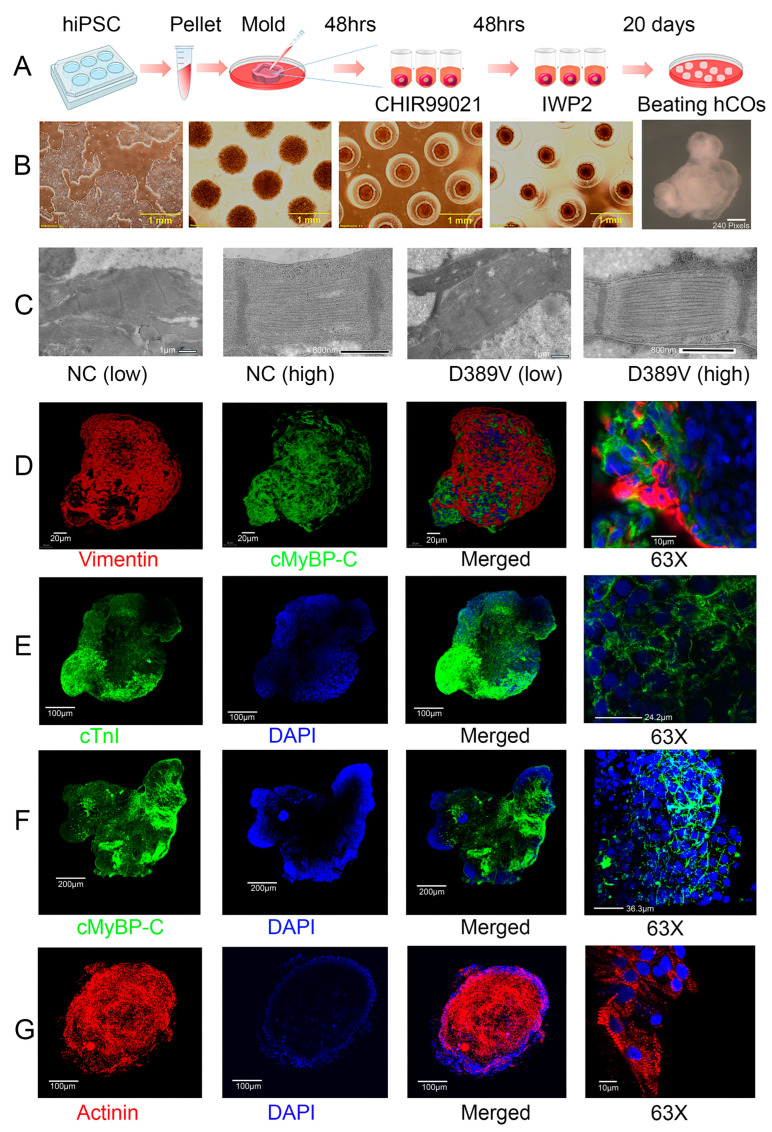
Generation and characterization of 3D cardiac organoids using hiPSC lines using Wnt signaling. (**A**,**B**) Cultured hiPSCs in a 6-well plate were seeded into the agarose mold for 48 h. The maintenance medium, mTESR1, was changed to RPMI 1640 containing B27 without insulin with the subsequent addition of CHIR99021 and then IWP4 to drive cardiomyocyte maturation. Organoids were transferred after 96 h to an ultra-low attachment plate with RPMI1640 medium supplemented with B27 plus insulin and changed every 48 h until beating. (**C**) Transmission electron microscopy analyses of hCOs on day 30 demonstrate representative sarcomeres with Z-bands in D389V and NC organoids. (**D**–**G**) Representative wholemount cardiac organoid stained with (**D**) vimentin in red, cMyBP-C in green, and DAPI in blue, (**E**) cTnI in green and DAPI in blue, (**F**) cMyBP-C in green and DAPI in blue, and (**G**) α-actinin staining of the sarcomere in red and DAPI in blue. The last row represents the organoid area with higher magnification at 63X.

**Figure 2 cells-13-01913-f002:**
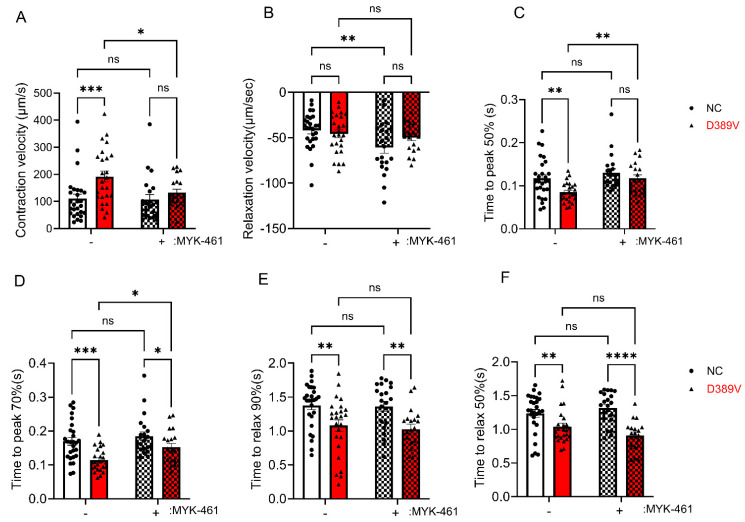
Hypercontractility was observed in D389V hCOs and was mitigated by mavacamten. Fresh beating hCOs on day 30 were used to measure contractile parameters using the IonOptix instrument. (**A**,**B**) Contraction and relaxation velocity in the absence and presence of 300 nM MyK-461 at 0.5 Hz frequency. (**C**,**D**) The time point taken to peak at 50% and 70% in the absence and presence of 300 nM MyK-461 at 0.5 Hz frequency. (**E**,**F**) Time to relax 90% and 50% was measured in the absence and presence of MyK-461 at 0.5 Hz frequency. The experiments were performed using n = 22 to 40 hCOs in 3 independent experiments on different days and averaged the data. Data are expressed as mean ± S.E.M. (error bars) of the number of hCOs. Statistical analyses in all groups were performed by ordinary two-way ANOVA, followed by a multiple-comparison test. For all groups: * *p* ≤ 0.05, ** *p* ≤ 0.01, *** *p* ≤ 0.001 and **** *p* ≤ 0.0001 and ns is non-significant.

**Figure 3 cells-13-01913-f003:**
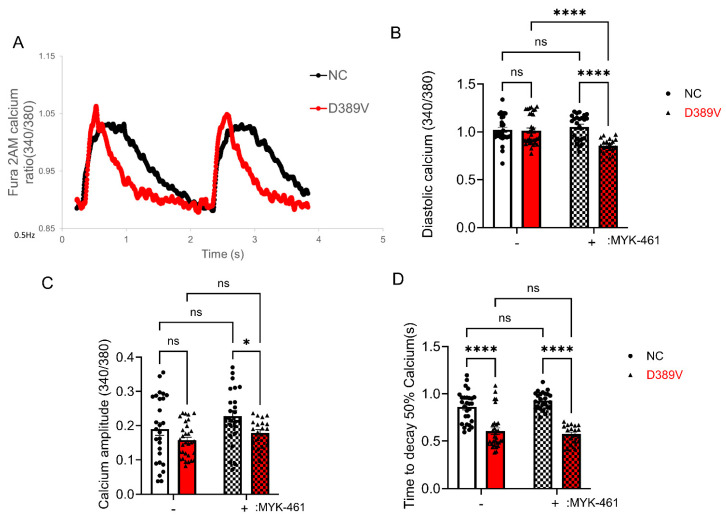
Faster Ca^2+^ kinetics were determined in D389V hCOs, which were not altered by MyK-461. Fresh beating hCOs on day 30 were used to determine the amplitude and kinetics of intracellular Ca^2+^ transients using fluorescent Fura-2 am dye. (**A**) Representative intracellular diastolic calcium levels tracings. (**B**) Intracellular diastolic calcium levels in D389V and NC hCOs with and without 300 nM Myk-461. (**C**) Ca^2+^ transient amplitude as indicated by Fura-2 ratio (340/380 nm) in the absence and presence of 300 nM MyK-461 at 0.5 Hz with 1.8 mm Ca^2+^. (**D**) Time to 50% decay of the calcium transient in the absence and presence of MyK-461. We used n = 22 to 40 hCOs in 3 independent experiments on different days and averaged the data. Data are expressed as mean ± S.E.M. (error bars), and statistical analyses were performed for all groups by ordinary two-way ANOVA, followed by multiple-comparison test. * *p* ≤ 0.05, and **** *p* ≤ 0.0001 for all groups, ns is non-significant.

**Figure 4 cells-13-01913-f004:**
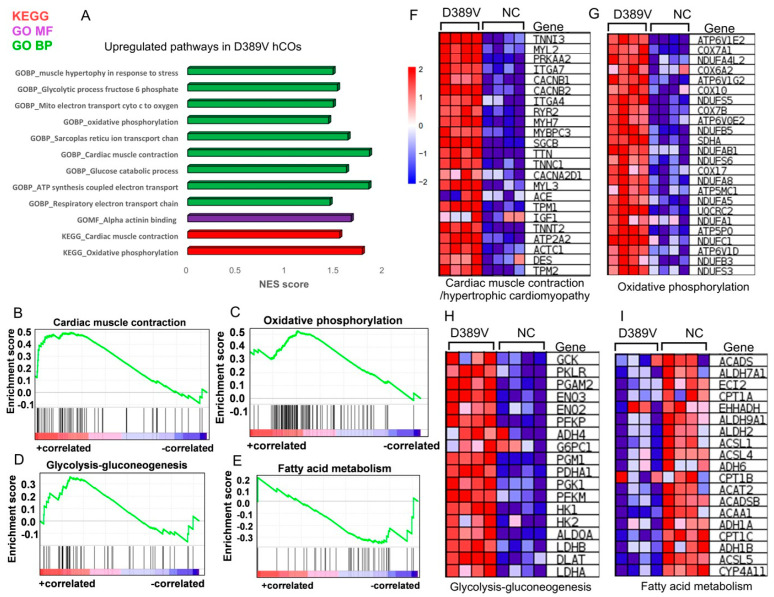
Expression of the D389V variant results in cellular hypertrophy and HCM phenotype, as evidenced by RNAseq analyses in hCOs. (**A**) Pathway enrichment analysis showing the KEGG, GOBP, and GOMF pathways for the upregulated genes in D389V hCOs compared to NC hCOs at day 30. (**B**–**E**) Enrichment plot using the GSEA analysis of the genes regulating the cardiac muscle contraction, HCM pathway, oxidative phosphorylation, glycolysis–gluconeogenesis, and fatty acid metabolism in D389V hCOs. (**F**–**I**) Heatmap representing DEGs associated with cardiac muscle contraction, HCM pathway, oxidative phosphorylation, glycolysis–gluconeogenesis, and fatty acid metabolism in D389V hCOs (n = 4 sample set with 25–30 organoids in each set; D389V vs. NC); (fold change cutoff 2, adjusted *p*-value < 0.05).

**Figure 5 cells-13-01913-f005:**
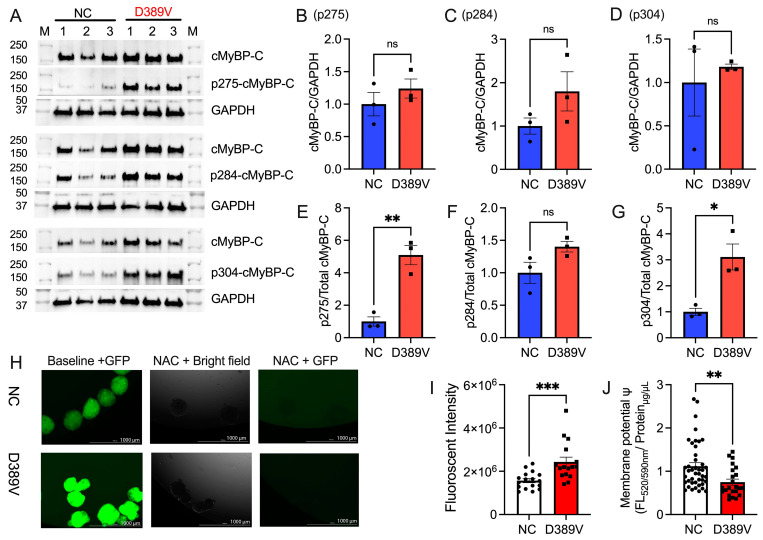
Expression of the D389V variant in hCOs increases the levels of cMyBP-C phosphorylation and oxidative stress. Total proteins were collected from hCOs on day 30 for Western blot analyses. (**A**) Representative of Western blot images and normalized protein expression levels comparing D389V and NC and hCOs. (**B**–**D**) Quantification of protein expression of cMyBP-C normalized with GAPDH for all three blots. (**E−G**) Quantification of the phosphorylation of human cMyBP-C at serine 275, 284, and 304 normalized with the level of total cMyBP-C protein. Data are expressed as mean ± S.E.M (error bars) on the number of organoids, and statistical analyses were performed in all groups *t*-test unpaired test. * *p* ≤ 0.05; ** *p* ≤ 0.01 for all groups. (**H**) DCFDA stain fluorescence at 485/535 nm of NC and D389V at baseline and after NAC treatment. Organoids imaged in brightfield microscopy (middle panel). All images are of scale 1000 µm and images at magnitude of 4X. (**I**) Quantitation of the ROS generation measurement of the fluorescent intensity emitted by the DCFDA dye measured in the hCOs. Data are expressed as mean ± S.E.M (error bars) (n = 3 sample sets, each sample set containing 40–50 organoids), and statistical analyses were performed in all groups *t*-test unpaired test. *** *p* ≤ 0.001 for all groups. (**J**) Quantification of the membrane potential, measured as the fluorescent intensity in the organoids. Data are expressed as mean ± S.E.M. (error bars) on the number of wells consisting of cells isolated from organoids, and statistical analyses were performed in all groups *t*-test unpaired test. ** *p* ≤ 0.01 for all groups. ns is non-significant.

**Figure 6 cells-13-01913-f006:**
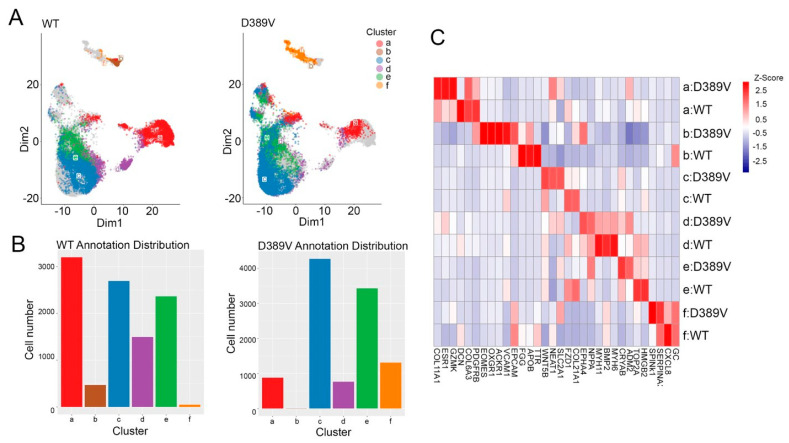
Spatial RNA detection determined all the cell types and cell–cell interactions in formalin-fixed paraffin-embedded hCOs. Fresh hCOs on day 30 were fixed in formalin, embedded in paraffin, and sectioned in 4–6 µm thickness for RNA profiling using a Human RNA TAP Panel (1000-plex) and SMI-0119 custom panel. (**A**) Uniform manifold approximation and projections of hCOs based on gene expression matrix in dimension 1 vs. dimension 2. Color denotes cell types from a to f types (a, a cluster of cells identified as a mixture of immune, endothelial, and fibroblast; b, macrophages; c, a cluster of cardiomyocytes, endothelial cells, and fibroblasts; d, cardiomyocytes; e, a cluster of cardiomyocytes; and f, immune cells). (**B**) Comparison of the gene expression levels in the same cluster of cells between D389V and NC hCOs in each cell type. (**C**) Heat map of gene expression profiles in each cluster in D389V and NC hCOs.

**Figure 7 cells-13-01913-f007:**
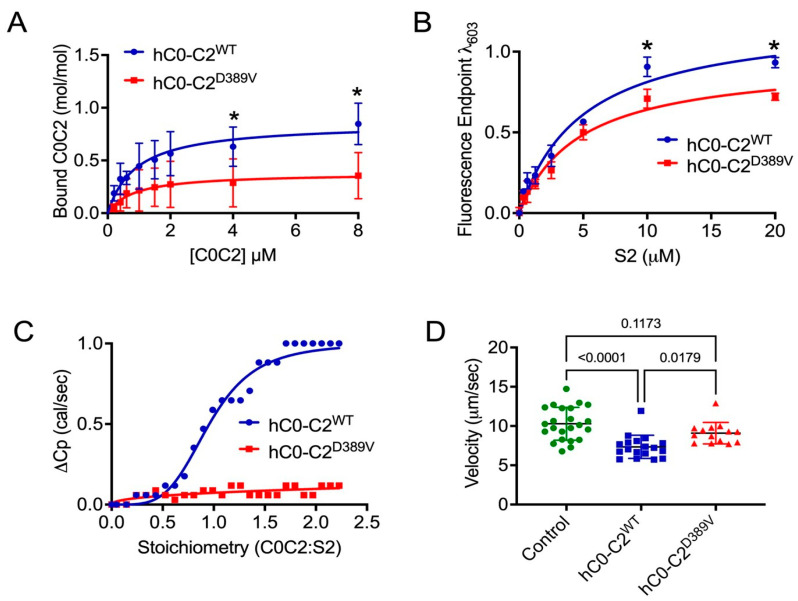
D389V mutation in the C2 domain of cMyBP-C reduces its binding to the myosin S2 region. (**A**) Co-sedimentation analysis of various recombinant hC0-C2 of cMyBP-C protein concentrations binding to full-length β-myosin heavy chains that were purified from mouse hearts. Quantification of bound hC0-C2 with myosin (y-axis) against utilized hC0-C2 (x-axis) in the assay was blotted for relative binding max (Bmax) and dissociation constant (*K*d). Data were fit to the Michaelis–Menten binding fit, and statistical analyses were performed using an unpaired *t*-test. Error bars indicate ± S.E.M. (* *p* < 0.05). (**B**) Solid-phase binding analysis between human myosin S2 (hS2) recombinant proteins and hC0-C2^WT^ as well as hS2 and hC0-C2^D389V^. Data were fit to the Michaelis–Menten binding fit, and statistical analyses were performed using an unpaired *t*-test to calculate the significance between the *K*d values. n = 3 with replicates of three for each n value. Data are expressed as the mean ± S.E.M. (**C**) Isothermal calorimetry analysis showing the sigmoidal curve for the titration of hS2 to hC0-C2 yields the dissociation constant (1/slope) and stoichiometry of the reaction. Both hC0-C2^WT^ and hC0-C2^D389V^ were titrated against 350 µM hS2 recombinant proteins. Statistical analyses were performed using one-way ANOVA with Tukey’s multiple-comparison test and single pooled variance (n = 3). Data are expressed as the mean ± S.E.M. (**D**) Scatterplot of in vitro motility assay to measure the average velocity of actin thin filaments over human myosin HMM thick filaments in the absence (green) and the presence of hC0-C2^WT^ (red) and hC0-C2^D389V^ (blue) proteins. Statistical analyses were performed using one-way ANOVA with Tukey’s multiple-comparison test and single pooled variance (n = 3). Data are expressed as the mean ± S.E.M. (* *p* < 0.05).

## Data Availability

The detailed description of experimental materials, methods, and data supporting this study is provided in the Appendix A. The high-throughput sequencing datasets generated from this study can be accessed through Gene Expression Omnibus accession numbers GSE262876 (RNAseq) and GSE263591 (Nanostring/spatial).

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
