# Peer review of "MYBPC3 D389V Variant Induces Hypercontractility in Cardiac Organoids"

_cells, 2024, doi:10.3390/cells13221913_

Round 1
Reviewer 1 Report
Comments and Suggestions for Authors
Dr Desai and collaborators used cardiac organoïds generated from human iPS cells derived from healthy individual or from patients harboring a polymorphic variant (D389V) in MYBPC3. Mutated MYBPC3 is a negative regulator of myosin ATPase and is responsible for a hypertrophic cardiomyopathy.
This study was designed to investigate the molecular mechanisms underlying the pathogenesis of HCM in D389V MYBPC3 organoïds.
This is a complete story in which the authors monitored biochemical, cellular as well as functional parameters.
I still have a few comments
1. D389V MYBPC3 leads to cardiomyocytes hypertrophy. It is mandatory to measure the size of cardiomyocytes in both wt and MYBPC3 mutated organoid. Cell enzymatic dissociation of organoïds to isolate cardiomyocytes may help in monitoring such a parameter.
2. A cardiomyocyte within the heart is submitted to a hemodynamic load. This is an important experimental condition to take into accounts wen ones investigates hypertrophic cardiomyopathy. I am not sure that the IonOptix device is the best system to use. Did the authors apply a load to cardiomyocytes in this set-up?
3. The RNA seq experiments points that D389V MYBPC3 myocytes still feature an embryonic glycolytic metabolism and not yes a fatty acid metbolism. Does it mean that they are less differentiated than wt? Would such a hypothesis fits with the expression of embryonic TnnI and Tnt isoforms?
4. Why is expression of MYBPC3 gene upregulated in MYBPC3 mutated organoids while the protein level is not changed? Is there a regulation at the translational level?
5. D389V MYBPC3 binds less myosin and in turn is unable to regulate myosin ATPase. This would explain the effect of the myosin inhibitor mavacamten. Is this correct or is there any other “side” effect of the drug? Indeed mavacamten decreased diastolic Cai.
6. The Cai experiments show that Ca increases more quickly and that it also more quickly decreases in the cytosol. Overall, it looks that cytosolic Cai is more transiently increased thus pointing to less activation of myosin ATPase? This would counteract the hypercontractilty?
Author Response
Response to the reviewers
Reference: Desai D et al.,
Manuscript ID: cells-3268000
Point-by-point response to the reviewers' comments:
We thank all the reviewers for their comments and suggestions, both of which have improved the presentation of our manuscript. We have addressed each reviewer's comments, and below is our point-by-point response to the concerns expressed. Our answers are in blue font below each critique. Additionally, any changes in the manuscript are highlighted using the track changes feature.
Reviewer 1
- D389V MYBPC3 leads to cardiomyocyte hypertrophy. It is mandatory to measure the size of cardiomyocytes in both WT and MYBPC3 mutated organoids. Cell enzymatic dissociation of organoids to isolate cardiomyocytes may help monitor such a parameter.
Answer: We agree with the concern and the suggested remedy, and we have tried many protocols using TryPLE (ThermoFisher Scientific) to digest cardiac organoids and dissolve cardiomyocytes to measure the cell size. However, each attempt was met with failure because organoids do not have perfusion valves; as a result, all cells died during the isolation procedure. Therefore, as a fallback, we used a previously published technique involving RNA-seq data to validate the presence of cardiac hypertrophic markers to validate the presence of cellular hypertrophy in 2D IPSC-CM in the presence of the D389V variant (Viswanathan SK, Puckelwartz MJ, Mehta A, et al. Association of Cardiomyopathy with MYBPC3 D389V and MYBPC3Δ25bpIntronic Deletion in South Asian Descendants. JAMA Cardiol. 2018;3(6):481–488. doi:10.1001/jamacardio.2018.0618). We have now added this point in the study limitation section in red (Page 11, last paragraph).
- A cardiomyocyte within the heart is submitted to a hemodynamic load. This is an important experimental condition to take into account when one investigates hypertrophic cardiomyopathy. I am not sure that the IonOptix device is the best system to use. Did the authors apply a load to cardiomyocytes in this set-up?
Answer: Thank you for the question. The cardiac organoid field is still emerging, so a number of techniques and assays still need to be established. In fact, we worked with the IonOptix team to establish measurements suitable for cardiac organoids. Contractility and calcium transients in isolated cardiomyocytes have been accurately measured by the IonOptix system for decades (PMID: 39196032). However, for organoids and iPSC-CMs, it is not feasible to measure sarcomere length shortening by using freshly isolated adult CMs. Therefore, IonOptix has created their Cytomotion software (Measuring iPSC-CM Calcium & Contractility - IonOptix) that leverages pixel movements within the area in cardiac organoids and iPSC-CMs to derive contractile kinetics and ratiometric quantitative calcium transients. Many studies have reported the use of software similar to Cytomotion, such as Musclemotion, with algorithms designed to measure contractile function (PMID: 29282212, PMID: 30253059, PMID: 37349842).
To remain unbiased, we did not take into account any cell shortening data as pixel movements do not clearly depict that. Instead, for our experiments, we considered the kinetics of contractility like contraction velocity, relaxation velocity, time taken to peak, and baseline. These data clearly point to the contractility of organoids and IPSC-CMs. Lastly, we applied mechanical stimulation (0.5Hz frequency at 5V) to both WT and mutant organoids, maintaining equal pace to determine the difference. Thus, we believe the present data are valid and reproducible. We did not apply any load to our cardiomyocytes because we were more interested in knowing the effect mediated by the D389V mutation at baseline, pacing all the organoids at the same frequency of 0.5hz and 5V. Thus, any frequency and voltage levels exceeding those specified resulted in unresponsive organoids. We have now added this information in the online methods section (Page 6).
- The RNA seq experiments points that D389V MYBPC3 myocytes still feature an embryonic glycolytic metabolism and not yes a fatty acid metbolism. Does it mean that they are less differentiated than wt? Would such a hypothesis fits with the expression of embryonic TnnI and Tnt isoforms?
Answer: Thank you for this comment. Indeed, when compared to adult cardiomyocytes, iPSC-CMs are still embryonic in nature and therefore feature glycolytic metabolism rather than fatty acid metabolism. Efforts are ongoing to differentiate them into more adult-like cardiomyocytes. Interestingly, in our 3D organoids vs 2D organoids, we do show better expression of cardiac and other sarcoplasmic reticulum genes in 3D organoids compared to 2D IPSC-CM (refer to Supplementary Figure 2).
To answer your next question, we would not classify D389V organoids as less differentiated than WT owing to the increase in glycolysis because cardiac metabolism in patients with HCM switches back to a more fetal phenotype characterized by an increase in glycolysis and a decrease in fatty acid β-oxidation (PMID: 20505524). Consequently, we believe our D389V organoids showed more glycolytic metabolism than fatty acid metabolism by early hypertrophic response.
The expression of TnI and TnnT isoforms is increased in D389V organoids compared to our WT organoids, indicating that the molecular hypertrophy results from the D389V mutation.
- Why is the expression of MYBPC3 gene upregulated in MYBPC3 mutated organoids while the protein level is not changed? Is there a regulation at the translational level?
Answer: Cardiac sarcomere proteins have a selective and specific translational process. At the RNA level, they can be upregulated or overexpressed in transgenic mice using cardiac-specific alpha-myosin heavy chain promoter. However, at the protein level, they always maintain 100% stoichiometry and are never overexpressed. We have previously reported these findings (Sadayappan, Circ Res. 2005 Nov 25;97(11):1156-63; Sadayappan, Proc Natl Acad Sci U S A. 2006).
- D389V MYBPC3 binds less myosin and in turn is unable to regulate myosin ATPase. This would explain the effect of the myosin inhibitor mavacamten. Is this correct or is there any other “side” effect of the drug? Indeed mavacamten decreased diastolic Cai.
Answer: We appreciate the comment. Owing to D389V mutation in the myosin binding region (Domain C2) of cMyBP-C, D389V MYBPC3 cannot bind myosin. This results in accelerating myosin to interact with actin to increase cross-bridges and significantly activate myosin ATPase, causing hypercontraction.
Mavacamten is also known to improve impaired relaxation during HCM (PMID: 34915982). During HCM, diastolic calcium increases in the cytosol and is unable to sequester back to SR efficiently. Mavacamten was shown to reduce diastolic calcium, thereby reducing calcium sensitivity, as described in this paper (PMID: 32083971). Therefore, the improved relaxation observed after mavacamten administration could result from reducing overall calcium sensitivity, thereby mitigating hypercontractility in HCM cardiomyocytes. Of course, mavacamten dosage should be titrated carefully as it inhibits myosin activities and reduces contraction, which explains why the drug cannot be prescribed to asymptomatic carriers as a prophylactic against the development of HCM.
- The Cai experiments show that Ca increases more quickly and that it also more quickly decreases in the cytosol. Overall, it looks that cytosolic Cai is more transiently increased thus pointing to less activation of myosin ATPase? This would counteract the hypercontractility?
Answer: Thank you for the insightful comment. Yes, in our mutant cardiac organoids, we observe no significant changes in calcium amplitude and diastolic calcium between NC and D389V cardiac organoids. However, we did see a faster calcium decay rate, indicating that the increased cellular calcium extrusion from the cytosol and faster uptake of calcium could be a compensatory response to hypercontractility. It is likely that this response occurs to ensure that sufficient calcium is available for the subsequent depolarization cycle. Interestingly, a similar phenomenon was also observed in myocytes from homozygous MYBPC3 C.772G>A mouse hearts, a mutation that also results in the replacement of a negatively charged amino acid (Glu258Lys) in which faster decay, but no significant change in amplitude, was observed (PMID: 26825555). Also, the function of cMyBP-C in regulating calcium sensitivity remains an unsettled question. This is evidenced by the varied outcomes observed in several cMyBP-C-knockout mouse models. For instance, Harris et al. (PMID: 11909824) reported a decrease in Ca2+ sensitivity in cMyBP-C-knockout mice, while Fraysse et al. (PMID: 22465693) found that Ca2+ sensitivity actually increased in another model lacking functional cMyBP-C. Conversely, Barefield et al. (PMID: 24464755) noted no significant change in Ca2+ sensitivity. Therefore, such lack of change in calcium amplitude observed in our D389V organoids suggests that the observed hypercontractility likely results from increased actomyosin interaction.
Reviewer 2 Report
Comments and Suggestions for Authors
In their study, the authors investigated the molecular mechanisms underlying the pathogenesis of HCM associated with D389V variant in MYBPC3 gene by using human-induced pluripotent stem cell-derived cardiac organoids.
I congratulate the authors for their thorough work. The manuscript flows smoothly and is clear. I have no major issues. Minor comments: please harmonize the magnification in Figure 1 D-G (64x) and in the legend (63x).
Author Response
Response to the reviewers
Reference: Desai D et al.,
Manuscript ID: cells-3268000
Point-by-point response to the reviewers' comments:
We thank all the reviewers for their comments and suggestions, both of which have improved the presentation of our manuscript. We have addressed each reviewer's comments, and below is our point-by-point response to the concerns expressed. Our answers are in blue font below each critique. Additionally, any changes in the manuscript are highlighted using the track changes feature.
Reviewer 2
Minor comments: please harmonize the magnification in Figure 1 D-G (64x) and in the legend (63x).
Answer: We thank the reviewer for this comment. We have now changed Figure 1 on page 16 with 63X